# Robust PPG Peak Detection Using Dilated Convolutional Neural Networks

**DOI:** 10.3390/s22166054

**Published:** 2022-08-13

**Authors:** Kianoosh Kazemi, Juho Laitala, Iman Azimi, Pasi Liljeberg, Amir M. Rahmani

**Affiliations:** 1Department of Computing, Faculty of Technology, University of Turku, 20014 Turku, Finland; 2Department of Computer Science, University of California, Irvine, CA 92697-3435, USA; 3Institute for Future Health, University of California, Irvine, CA 92697, USA; 4School of Nursing, University of California, Irvine, CA 92697, USA

**Keywords:** PPG, peak detection, convolutional neural network, wearable devices, motion artifacts

## Abstract

Accurate peak determination from noise-corrupted photoplethysmogram (PPG) signal is the basis for further analysis of physiological quantities such as heart rate. Conventional methods are designed for noise-free PPG signals and are insufficient for PPG signals with low signal-to-noise ratio (SNR). This paper focuses on enhancing PPG noise-resiliency and proposes a robust peak detection algorithm for PPG signals distorted due to noise and motion artifact. Our algorithm is based on convolutional neural networks (CNNs) with dilated convolutions. We train and evaluate the proposed method using a dataset collected via smartwatches under free-living conditions in a home-based health monitoring application. A data generator is also developed to produce noisy PPG data used for model training and evaluation. The method performance is compared against other state-of-the-art methods and is tested with SNRs ranging from 0 to 45 dB. Our method outperforms the existing adaptive threshold, transform-based, and machine learning methods. The proposed method shows overall precision, recall, and F1-score of 82%, 80%, and 81% in all the SNR ranges. In contrast, the best results obtained by the existing methods are 78%, 80%, and 79%. The proposed method proves to be accurate for detecting PPG peaks even in the presence of noise.

## 1. Introduction

There is a growing demand for ubiquitous health monitoring systems. These systems are developed to provide proactive healthcare solutions as well as reduce medical costs: e.g., providing efficiency and cost-savings for doctors, nurses, and pharmaceutical companies [1]. Fortunately, rapid advancements in the Internet of Things (IoT)-based systems and wearable devices offer opportunities for the development of health monitoring systems [2]. Such IoT-based healthcare systems can provide comprehensive patient care by leveraging various sensor types, communication units, and computing resources. Wearable electronics—such as wristbands and smart rings—enable the ubiquitous collection of biomedical signals, including electrocardiogram (ECG) and photoplethysmogram (PPG) [3].

PPG is a low-cost, non-invasive, and simple optical technique used for measuring the synchronous blood volume changes in tissue such as the surface of the finger, toe, wrist, and forehead [3]. This approach is widely used in wearable IoT-based applications due to its high level of feasibility and ease of measurement [4]. Collected PPG signals can be used to extract various health parameters, such as heart rate and heart rate variability. These health parameters are obtained by the determination of the systolic peaks in the PPG records. However, the quality of the PPG waveform is easily affected by surrounding noises such as background noises and motion artifacts. The noises are unavoidable in IoT-based healthcare systems, as users engage in a variety of physical activities. Subsequently, when the signal quality is poor (i.e., low signal-to-noise ratio (SNR)), accurate detection of peaks in PPG signals becomes challenging. This issue increases false peak determination, which results in inaccurate vital signs extraction.

Numerous studies have been proposed to determine the PPG signal peaks accurately. In traditional methods, signals are inspected by experts, and then the peaks’ locations are annotated manually. These methods are often implemented in hospitals/clinics and are mainly used as gold standard methods for validation [5]. Such manual peak detection methods are time-consuming and require domain knowledge. Therefore, they are not feasible in health monitoring applications due to the growth of data volume over time.

On the other hand, various automatic signal processing techniques have been developed for PPG peak detection. These methods mainly include adaptive threshold [6], transform-based techniques [7], derivative calculation [8], and computer-based filtering [9]. Adaptive threshold techniques are commonly used for peak detection in biomedical signals (e.g., ECG and PPG). The methods set a threshold, which is increased or decreased adaptively based on the amplitudes of detected peaks in the signals. The threshold can be updated according to different attributes, such as duration, amplitude, beat-to-beat intervals, and sampling frequency [10,11]. In some cases, the threshold-based methods require ECG signals in addition to PPG signals, which add to the cost of the medical equipment and hinder their use in (remote) healthcare systems [12].

In addition, transform-based techniques are proposed for PPG peak detection. These methods leverage signal processing techniques, such as discrete wavelet transform [13], stationary wavelet transform [7], and Hilbert transform [14,15]. Wavelet-based techniques are mostly used for denoising. These techniques decompose signals into multiple sub-bands with the same resolution as the original signals. Then, by composing the desired sub-bands, the informative parts are regenerated, and baseline wanders and high-frequency (HF) noises are eliminated. Hilbert transform is also employed for peak detection tasks. Hilbert transform is a powerful tool in analyzing the amplitude and frequency of a signal instantaneously. Refs. [14,16] indicate that the zero-crossing points in the Hilbert transform correspond to peaks’ locations. However, these methods are insufficient for noise-contaminated signals, and they become unreliable if the SNR drops below a certain level.

In addition, machine-learning-based approaches have been developed for PPG signal analysis [17,18]. For example, a three-layered feedforward neural network was introduced in [19] for PPG peak detection. The method was only trained and evaluated with low-noise signals.

The conventional peak detection techniques in the literature are mainly designed for noise-free or low-noise PPG signals. Therefore, they are insufficient to determine PPG peaks’ locations when the signal quality is poor due to motion artifacts and HF noises. These noises are inevitable in wearable-based health and well-being monitoring systems. We believe that a peak detection method is required to determine systolic peaks in noisy PPG, leveraging temporal information in the signal. The robustness of such a method requires to be investigated against different noise levels.

In this paper, we propose a CNN-based peak determination approach for PPG signals with different levels of motion artifacts. The convolution layers in our network are dilated, resulting in a large receptive field. Therefore, the network can use temporal information in PPG peak detection and learn complex problems associated with the noisy PPG signals. Our analysis exploits PPG signals and motion artifacts collected by wearable devices in health monitoring under free-living conditions. We develop a generator function to produce PPG signals with a wide range of noise, augmenting the training data and creating noisy signals similar to real-life PPG records. Using the PPG signals, the proposed method is evaluated in comparison with state-of-the-art PPG peak detection methods. In summary, the major contributions of the paper are as follows:Proposing a dilated convolutional-neural-networks-based method for addressing the problem of PPG peak detection in the presence of noise.Assessing the robustness of the proposed method using noisy PPG signals with SNRs ranging from 0 to 45 dB.Evaluating the proposed method in terms of accuracy compared to conventional methods, including adaptive threshold and Hilbert transform.Providing the model implemented in Python for the community to be used in their solutions (https://github.com/HealthSciTech/Robust_PPG_PD).

The rest of the paper is organized as follows. The background and related work of this research is outlined in Section 2 and Section 3. We introduce the dataset used in this work in Section 4. Section 5 describes the development of the proposed method in detail. In Section 6, we evaluate our method in comparison with other published methods. Finally, Section 7 concludes the paper.

## 2. Background

In this section, we briefly describe PPG and neural networks proposed for PPG-based applications.

### 2.1. Photoplethysmography

PPG is a convenient method for sensing the blood flow rate at peripheral sites. Therefore, this signal can be used to determine the cardiac cycle [3]. The PPG sensor includes two main components, i.e., a light source and a photodetector. PPG signals are acquired by emitting light in different wavelengths (e.g., infrared, red, and green, often at 940, 660, and 550 nanometers, respectively) to the skin surface and capturing the reflected light via photodetectors. The infrared and red lights are commonly used for measuring heart rate and blood oxygen saturation. Furthermore, the green light is widely used in wearable devices such as smartwatches [20].

The variation in the PPG signal is associated with cardiac and respiration oscillations. Figure 1 indicates a view of a PPG signal, where the heart rate values can be estimated by measuring the difference of the time interval between two successive peaks. The signal consists of two main components, i.e., the alternating current (AC) and direct current (DC). The AC part denotes synchronous cardiovascular fluctuations caused by cardiac activity, while the DC portion denotes various low-frequency elements of the blood flow, such as respiration [3,18].

The PPG method is widely used in wearable and mobile applications [20]. However, the collected signal is highly susceptible to noise. The typical PPG noises include motion artifact, baseline wander, and environmental noise [21]. Motion artifacts are generated due to the user’s hand movements. Baseline wander is a low-frequency noise, which is noticeable as a fluctuating pattern in the PPG signal. Environmental noise is produced by additional sources (e.g., ambient light) collected in addition to PPG during the monitoring. The noise level in the PPG signal depends on different factors, such as the sensor quality, sensor setup (e.g., electric current), intensity of the activity, and environmental factors [20,22]. In this study, the additive noise includes baseline wander and motion artifacts, and we focus on the noise levels in PPG signals.

### 2.2. Neural Networks in PPG Applications

Artificial neural networks are inspired by the human brain and imitate how biological neurons interact with one another, comprising an input layer, hidden layers, and an output layer [23]. Neural networks algorithms have been recently used in various PPG signal applications. In [24], a classification method based on a multilayer perceptron (MLP) network was presented. In their study, an MLP network was trained to classify the pattern of the onset and systolic of the PPG signals with different window sizes. The preprocessing stage includes two steps, i.e., signals segmentation and smoothing using a simple mean square regression. Then, the results are fed to the network as features for pattern recognition. Chen et al. [25] proposed a hidden Markov model for PPG classification. They first used linear predictive coding and sample entropy methods to extract different features from the PPG waveforms. Then, a vector quantization method was employed to convert the features into the prototype vectors which were utilized to estimate the parameters for hidden Markov model parameters. Reiss et al. [26] introduced a CNN architecture for heart rate estimation. In their study, the PPG signals and corresponding three-axis accelerometer data were used to train the model.

For PPG noise removal, Ref. [17] proposed a deep recurrent neural network and stochastic modeling recover the noise-corrupted PPG signals. They first used recurrent neural networks for segmentation. Then, a Kalman filter was employed to extract clean PPG and create a stochastic model. They also tested their method on a real-time dataset acquired by a wearable glove. In addition to noise removal, deep learning methods were proposed for PPG quality assessment [18,27]. In these studies, 1D and 2D CNN models were trained to discriminate between reliable and unreliable signals. The methods were evaluated by comparing the result with ECG references.

## 3. Related Work

In this section, we describe several PPG peak detection methods with different complexities which have been developed in the last decades. Most peak detection methods contain two main stages, i.e., Section 3.1: preprocessing (or filtering) and Section 3.2: envelope detection and peak determination.

### 3.1. Preprocessing

Preprocessing is one of the important stages in the PPG peak detection task. This step aims to remove components of the signal that do not reflect the features of interest (e.g., heart rate and HRV). In the preprocessing step, different filtering methods—such as low-pass filter, high-pass filter, adaptive filter, singular value decomposition, and mode decomposition—are employed to suppress the baseline distortion and HF noises. Such methods make the systolic peak part more prominent in the PPG signals. Ref. [28] proposed low-pass and high-pass filtering methods with cut-off frequencies of 0.4 and 8 Hz to remove motion artifact and HF noise, respectively. The significant component of background noise is presented in the frequency range of 0.15 to 5.0 Hz. This was achieved by employing a band-pass filtering method with cut-off frequencies of 0.5 and 5.5 Hz [29]. Prieto et al. [30] used a combination of two zero-phase delay fourth-order high-pass and eighth-order low-pass Butterworth filters with a bandwidth of 0.1–16 Hz to remove unwanted signals. Moving average filters were also utilized in noise surpassing. For instance, in [8], a three-point bidirectional moving average was proposed to remove the phase delay caused by the filter. In [31], the authors proposed a novel two-step method consisting of noise filtering and noise elimination. In the noise filtering step, a band-pass filter was followed by a three-point moving average filter for signal smoothing. The signal was segmented into cycles for the noise elimination part, and three statistical features (i.e., standard deviation, kurtosis, and skewness) were calculated. Then, a threshold was set based on the features, and the segments beyond the threshold were eliminated. A three-point moving average filter was used for smoothing the signal at the final stage.

In [32], a variational mode decomposition was used to enhance the signal quality and suppress the motion artifact. The decomposition was implemented in two stages to minimize the balancing error. Paradkar et al. [7] introduced a singular value decomposition along with a moving average filter to extract the periodic component from the raw signal and reduce background noise. Ye et al. [33] proposed a hybrid motion artifact removal method combining adaptive filtering and signal decomposition. In this method, the PPG records and the acceleration data (used as the reference) were used as the input of the adaptive filter simultaneously. After using the adaptive filter, a signal decomposition method was performed to remove remaining motion artifact components of the raw PPG signal. Similarly, an adaptive filter was introduced for motion artifact reduction in [34]. The PPG-based filtering techniques reduce the effect of noise, but they cannot guarantee highly accurate results all the time. For example, they cannot remove noise peaks, when the noise overlaps (e.g., in frequency) with the PPG peaks. Moreover, they might require additional sensors, e.g., accelerometer or gyroscope.

### 3.2. Envelope Extraction and Peak Determination

This stage generally includes extracting different features such as the maxima and minima, slope of the signal, and signals’ envelope using well-known and robust algorithms, e.g., adaptive threshold, transform-based, and machine-learning-based algorithms to determine the signal peaks.

#### 3.2.1. Adaptive Threshold

The adaptive threshold is a common technique used for PPG peak detection. This method employs a constant specified by signals’ temporal and frequency domain features and time intervals. The constant could be decaying or growing due to the dynamic nature of the PPG waveform [6,28]. For example, Shin et al. [35] proposed to update the threshold according to different features such as the sampling frequency, preceding peaks, and standard deviation of the signal. In [10], the adaptive thresholding is equipped with a morphological filter to remove the low noises and a slope sum function is used to pinpoint the location of peaks accurately. Van Gent [36,37] presented a method based on an adaptive threshold followed by moving average and spline interpolation methods if the detected peaks show clipping. In [38], a derivative-based method was proposed equipped with a slope sum function and an adaptive threshold to mitigate false peak detection. The complexity of the adaptive threshold PPG peak detection methods is low. However, the methods are sensitive to noise and fail to accurately identify the peaks when the PPG signal is contaminated by noise. In other words, the PPG signal changes rapidly due to noise, so the methods are incapable of selecting appropriate thresholds.

#### 3.2.2. Transform-Based Techniques

In addition, transform-based techniques are developed for PPG peak detection. These methods are mainly based on various linear transformations such as wavelet [7] and Hilbert transforms [14], by which the signal’s temporal and frequency domain features are extracted. Then, various thresholds—such as zero-crossing points or decision logic—are set to extract the signals component and corresponding peaks in the original signals. In [14], a Hilbert transform was accompanied by moving average and Shannon energy envelope techniques to locate the position of the systolic peaks. In [15], a Hilbert double envelope method was proposed for PPG peak detection. Through Hilbert transform, the lower and upper signal envelopes were obtained. In the next step, the Hilbert transform was applied to the upper envelope, and the lower envelope was retrieved. Finally, a local maximum method was applied on the output signal to locate the PPG peaks. Vadrevu et al. [39] introduced a stationary wavelet transform to extract two sets of coefficients from the PPG signal. Then, using multiscale sum and product, the peaks’ sharpness was enhanced in the edges, and the other values remained near zero. Following that, the zero-crossing points were extracted to obtain the locations of the systolic peaks. Leveraging transform-based methods, the peak positions can be detected more accurately. For instance, Ref. [40] proposed a robust algorithm—enabled by a Hilbert transform, amplitude thresholding, and signal derivative—to detect PPG systolic peaks. Their algorithm achieved a better performance in comparison to an adaptive threshold technique. However, the transform-based methods are still insufficient for wearable-based PPG, as they fail to determine systolic peaks in distorted PPG signals precisely.

In another work, Ref. [11] introduced a positioning algorithm to locate PPG peaks. The method included denoising and abnormal intervals removal steps. In [41], PPG peaks were automatically detected and corrected, exploiting a Poincare plot feature and envelope detection. In [38], a window-based approach was introduced for peak detection. The peak location was determined by sliding a certain window over PPG signal. Following that, several well-documented, time-domain strategies (i.e., refractory period, clipping detection, and peak verification) were proposed to address disturbances associated with PPG signals. However, the method’s accuracy is highly affected by the window size. The methods mentioned above are accurate when the PPG signal quality is good. However, they are highly susceptible to motion artifacts and environmental noises. They fail to differentiate false noisy peaks and systolic peaks and subsequently result in inaccurate peak detection. Moreover, the probability of false peak detection increases in signals with a high heart rate. Consequently, these methods are insufficient for wearable-based monitoring, in which the users might engage in various physical activities.

#### 3.2.3. Machine Learning Methods

Traditional machine learning and deep learning methods have been employed to analyze cyclostationary biosignals such as PPG and ECG. For example, Ref. [42] proposed 1D CNN for QRS complex detection. In the preprocessing stage, a derivative function followed by an averaging system was used for noise removal. Then, the signals were fed to the CNN method for automatic feature extraction and classification. In [43], a faster R CNN model was proposed for ECG peak detection. Their method included three steps. First, the ECG signals were segmented and transformed into 2D images. Second, the images were fed to the model, and the output features map was input into a regional proposal network. In the final step, by setting a threshold, low probability outputs were excluded. Their method was tested with 24 h wearable ECG recordings. In addition, Ref. [44] proposed an automatic R-peak detection for ECG signals. The method comprised a bidirectional LSTM to obtain the probabilities and locations of R-peaks. The machine-learning-based methods mentioned above have been utilized for ECG R-peak detection. They are insufficient for PPG signals due to the difference in the signals’ origins. PPG systolic peaks detection is more challenging as the peaks’ slopes are not as large as QRS-complex in ECG. PPG signal quality and the waveform are also highly susceptible to artifacts generated, for example, by the user’s hand movements.

For PPG peak detection, Ref. [19] proposed an online sequential learning algorithm. Their method included two steps. First, they divided the PPG signals into a set of fundamental sinusoidal defined segments. Among these segments, only one segment contained a peak. In the second step, a feedforward neural network method was trained to detect peaks in the segments. However, the method could not differentiate systolic peaks from noise peaks so it might fail with noisy signals. Moreover, the evaluation was merely limited to noise-free and low-noise PPG signals.

## 4. Dataset

The PPG dataset used in this paper is a part of a health monitoring study [45]. During the study, the participants were asked to wear Samsung Gear Sport smartwatches, with which their vital signs, physical activity, and sleep were tracked continuously. The monitoring was performed under free-living conditions, where the participants engaged in their normal daily routines.

The recruitment and data collection took place in southern Finland between July and August 2019. The recruitment started with the students and staff members of the University of Turku. More recruiting was then performed with snowball sampling, and in the end, 46 individuals were recruited. All of the participants were healthy individuals, and both males and females were present in equal numbers. The following exclusion criteria were used in the recruitment: (1) any restrictions using wearable devices at work, (2) restrictions regarding physical activity, (3) a diagnosed cardiovascular disease, and (4) symptoms of illness at recruitment time. Due to technical and practical issues, PPG signals from all 46 participants were not available, and data from 10 participants had to be excluded. Thus, PPG data from 36 participants were used in our analysis. Table 1 summarizes the background information of the participants.

All PPG signals were recorded with Samsung Gear Sport smartwatches [46]. The smartwatch has compact dimensions of 44.6 × 42.9 × 11.6 mm, and it weighs 67 g with the strap. The smartwatch is waterproof, its battery lasts about 3 days, and it includes a PPG sensor and a built-in inertial measurement unit. The device runs an open-source Tizen operating system, enabling customized data collection and data transmission.

For the data collection, the participants were asked to wear the smartwatches on their non-dominant hands. The watches were programmed to collect data for 24 h at the sampling frequency of 20 Hz. We upsampled the PPG signals to 100 Hz to include the tolerance distance in the peak detection (see Section 5.3.2). The upsampling was performed using a linear interpolation technique, which is a conventional method for upsampling low-frequency signals [47]. In this method, a line is fitted between each pair of data points. Then, based on the upsampling rate, new data points are fitted on the line. The participants were also asked to send the collected data via Wi-Fi to our server using our Tizen app [45]. Our monitoring system is depicted in Figure 2, including the Samsung Gear Sport smartwatch for data collection, a smartphone as a gateway layer for data transmission, and the cloud server.

This study was conducted following the ethical principles set by the Declaration of Helsinki and the Finnish Medical Research Act (No 488/1999). In addition, the University of Turku Ethics committee for Human Sciences gave a favorable statement (No 44/2019) of the study protocol. All study participants received both oral and written information about the study before their written consent was obtained. Study participation was entirely voluntary, and at all times, the participant had a right to withdraw from the study without giving any reason. At the end of the monitoring period, each participant was compensated with an EUR 20 gift card.

## 5. Deep Learning-Based PPG Peak Detection

In this section, we present a deep-learning-based method designed for PPG peak detection. The method is trained using noisy PPG signals. In the following, we first describe the data preparation step, including a generator function to produce noisy PPG signals. We then present the proposed model architecture and peak extraction method. The data analysis pipeline is shown in Figure 3.

### 5.1. Data Preparation

Data preparation generates noisy PPG signals with different SNRs to train and test the proposed model. The signals are generated using the available database, presented in Section 4. In this regard, we extract clean PPG signals and noise from the database. The collected PPG signals are nonstationary in terms of the noise level. In other words, the noise levels vary throughout the monitoring due to, for example, the user’s hand movement. Hence, the signals are divided into (quasi)stationary segments, within which we assume the noise level is fixed. The length of the segments should be long enough to allow meaningful waveform analysis but short enough to ensure the segments are (quasi)stationary. Note that too-short segments lead to low-resolution features. In our analysis, 15 s segments are selected.

The clean PPG signals are obtained using a PPG quality assessment technique, including five morphological features (i.e., spectral entropy, Shannon entropy, approximate entropy, kurtosis, and skewness) [48] and a support vector machine method. Then, the peak annotations (of the clean signals) are performed using a derivative-based method. In addition, we verify the method’s accuracy by randomly selecting the annotation outputs and examining them manually.

Moreover, baseline wander and motion artifacts are extracted from the collected data and stored in the noise dataset. The noises are additive and independent. Then, the clean signals (along with the peak locations) and noises are fed to a generator function.

#### Generator Function

A generator function is designed to create noisy PPG signals by aggregating clean PPG with noise. The noisy PPG signals are then utilized for training and testing the model. The generator function returns batches of normalized noisy PPG signals, their SNR values, and systolic peaks labels. Figure 4a shows a view of a generated PPG signal and its labeling vector.

The generator function includes five steps, as follows:

**Clean PPG signal selection:** A 15 s window of clean PPG signal (*X*) is randomly selected.

**Noise selection:** A 15 s window of noise (*N*) is randomly selected. Note that our noise dataset includes baseline wander and motion artifacts.

**Noisy PPG generation:** A weighted arithmetic mean is utilized to create the noisy PPG signals:(1)$$S=w_{X}X+w_{N}N$$
where $w_{X}$ and $w_{N}$ are the weights of the clean PPG signal and noise, respectively. In our case, $w_{X}$ is 1 while $w_{N}$ is a random number with uniform distribution (0, 5). Therefore, PPG signals with different noise levels are constructed. Then, the signals are normalized to [−1, 1] to be used for training and testing our model.

**Labels extraction:** A binary format is used for labeling the systolic peaks in the constructed PPG signal. In this labeling, “1” corresponds to the peak locations, whereas the rest of the signal is labeled as “0”. Moreover, a slightly balanced “1” is added to the adjacent systolic peak points for making the model more robust against the false positive. In other words, despite considering one point as the location of the peaks, five labels (i.e., peak, two preceding, and two succeeding points) are set to “1” (see Figure 4). The use of five “1”s instead of only one “1” in the labeling vector leads to more robust positive predictions. Therefore, it reduces the noise effect in identifying the peak’s location. It should be noted that the label values are created according to the systolic peaks in the clean PPG signals (but not in the aggregated noisy PPG).

**SNR extraction**: The SNR is calculated for each constructed noisy PPG signal as follows:(2)$$SNR=10\log\frac{P_{Signal}}{P_{Noise}}$$
where $P_{Signal}$ and $P_{Noise}$ are the signal and noise powers, respectively. The procedure of the generator function is also indicated in Algorithm 1.
**Algorithm 1** The generator function.**Initialize**:                win_size ← window size                batch_size ← number of batch size                $w_{X}$← 1**while** i< batch_size **do**   *X*← select a window of the clean PPG signal randomly   clean peak ← Extract the corresponding peaks locations   *N*← select a random noise with same window size   $w_{N}$← a random number with uniform distribution (0, 5)   *S*←$w_{X}X+w_{N}N$   norm_sig ← normalize the noisy signal (i.e., S)   label ← create a binary label format for the noisy signal   SNR ← calculate the SNR   *i*+=1**end while**

### 5.2. Model Architecture

To detect PPG peaks, we develop a CNN architecture with dilated convolutions, also known as atrous convolutions (or convolution with holes). Using dilated convolution instead of the regular one will result in a larger receptive field with the same amount of trainable parameters. This is achieved by inserting holes into the filter, i.e., some of the inputs are skipped as indicated in Figure 5. Dilation rate controls the amount of skipping, and filters with higher dilation rates have more holes. Dilated convolution with a dilation rate of 1 is a particular case that equals to a standard convolution. Dilated convolutions were utilized for the first time in efficient wavelet decomposition [49]. Later, they were successfully utilized in different deep learning applications, such as semantic image segmentation [50,51] and audio generation [52].

In our CNN model architecture, dilated convolutional layers are stacked, and their dilation rate is doubled at every layer. This approach results in vast receptive fields even with few layers (see Figure 6), which is computationally very efficient. Moreover, the input resolution is retained through the network. In contrast with our method, other existing methods that expand the receptive field, such as strided convolutions (stride larger than 1) or pooling layers, reduce the spatial resolution [53]. Stacking dilated causal convolutional layers and simultaneously increasing dilation rate was first proposed by Oord et al. [52] as part of their wavenet architecture for audio generation. Our model is architecturally simpler as we use a feedforward structure without any residual or skip connections. We also do not enforce causality. Therefore, receptive fields of the neurons in our model can contain both preceding and succeeding information, which will allow our model to make more accurate predictions.

Our model is fully convolutional, and it is a stack of seven 1D convolutional layers, as indicated in Figure 7. The input resolution of 1500 time steps is retained through the model. The kernel size is also 3 for every layer. The model makes sequence-to-sequence mapping. It produces a probability value for every time step. The probability value indicates how likely a signal point is to be a systolic peak. Two PPG examples with the probability values (i.e., the CNN model predictions) are shown in Figure 8. The dilatation rate is 1 in the first convolutional layer, and it is doubled at every following layer, reaching 64 at the final convolutional layer. This network structure results in a wide receptive field of 255 time steps for a neuron in a final classification layer. The size of the receptive field *r* of any convolutional layer *l* in our model can be determined as shown below:(3)$$r_{l}=2^{l+2}-1\;\;\mathrm{for}\;\;l=0,1,\ldots ,6$$

To keep our model compact, we slowly increase the number of filters as the network becomes deeper. The first convolutional layer contains four filters, while the second to last convolutional layer contains 32 filters. The final convolutional layer performs the binary classification; therefore, it has only one filter. It uses the sigmoid function (Equation (Equation 4)) as the activation function, while all preceding layers use exponential linear unit (Equation (Equation 5)) as the activation function [54].
(4)$$f\left(x\right)=\frac{1}{1+e^{-x}}$$
(5)$$f\left(x\right)=\left\{\begin{array}{cc} x & x>0,\\ \alpha (e^{x}-1) & x\leq 0 \end{array}\right.$$
where α > 0. Moreover, we chose Adam [55] as the optimizer and the binary cross-entropy as the loss function. The binary cross-entropy loss *L* is defined as follows:(6)L(y,p)=−(ylog(p)+(1−y)log(1−p))
where *y* is a binary class label (0 or 1) and *p* is the predicted probability indicating how likely it is that the prediction belongs to the positive class labeled as 1. The proposed model is very small since it only has 3169 trainable parameters.

### 5.3. Peak Finder

We develop a wrapper function to extract the locations of the precise peaks from the model predictions, provided by the CNN model. Moreover, the wrapper function detects and removes false peaks from the model predictions. Mainly, the function performs the three following tasks:Removes the peaks with low-probability predictions.Extracts the precise peak locations within the predicted values.Discards false peaks in the predictions.

In the following, we describe the three tasks of the wrapper function in more detail.

#### 5.3.1. Low-Probability Signal Removal

In the first step, predictions with low probability are discarded using a threshold value. A labeling vector—where each time step indicates one probability value between 0 and 1—is fed to the wrapper function. Then, a local threshold filter is applied to the predicted time steps, and the time steps below the defined probability value are filtered out. The threshold is chosen empirically after a considerable number of predicted time steps evaluation. Decreasing the probability threshold improves the recall but reduces the precision.

#### 5.3.2. Peak Extraction

We use a local maximum finder to determine the exact peak’s location. For this purpose, we design a searching function to find the five samples segment that has the higher value of the probability within the model predictions. In each five samples segment of the predicted time steps, the index of the higher probability is chosen as the location of the peak. Moreover, if there are two same probability values in the selected segment, the first probability value is chosen, and the corresponding index is extracted as the location of the peak. Figure 9 illustrates a segment of model prediction with its corresponding probability values. As shown in Figure 9, seven samples are above the threshold. In this example, the function finds the sample with the highest probability (i.e., 0.85) by comparing the neighbor points.

We produce a balance labeling vector for the noisy PPG signals, as instructed in Section 5. This idea helps our model to achieve higher precision while maintaining a lower tolerance distance. In other words, in the data generation stage, we introduce a new method for labeling. The method was generating a series of five “1”s instead of only one “1” as the location of the peak. This means that if the algorithm finds a peak in the peak detection phase, there might be a time difference between the exact peak location and the detected peak. This time difference is introduced as tolerance distance. Figure 10 shows a segment of the PPG signal and the defined tolerance distance with gray shaded rectangulars. In the proposed method, the tolerance distance is 50 ms, which is smaller than the tolerance distance (i.e., 88 ms) in other studies in the literature [44,56].

#### 5.3.3. Peak Correction

In the third step, too-close peaks are discarded. Ventricular depolarization cannot occur in the refractory period despite the presence of stimuli. Therefore, no peak is presented in PPG signals during the refractory period after a peak. Our analysis assumes that the maximum heart rate is 200 beats per minute and, accordingly, the minimum distance between two successive peaks is 300 ms. This step is necessary when we aim to maximize the recall while a low-value probability threshold is defined.

Accordingly, the PPG peaks within a distance less than the threshold (i.e., 300 ms) are considered as false peaks. In this regard, we add a peak into the false-peak list if the distance with its preceding peak is less than 300 ms. Then, the false-peak list is sorted based on the peak’s probabilities. In the next step, we select the highest peak’s probability in the false-peak list and add it to the peak list. Then, we calculate the distance with its preceding peak. If the distance would be larger than the threshold, it is chosen as a peak; otherwise, it is removed. We repeat this step until the false-peak list is empty. For clarity, let us take an example of the PPG peak correction. Four systolic peaks are indicated in Figure 11. In the first round, we calculate the distance between each peak with its preceding peak. As shown in the figure, the distance between the first peak ($P_{1}$) and the second peak ($P_{2}$) is 250 ms. Therefore, $P_{2}$ is added to the false-peak list. Likewise, $P_{3}$ is added to the false-peak list. In the next step, we sort these false peaks based on their probabilities. Then, we start with the highest probability (i.e., $P_{3}$) and calculate its distance with $P_{1}$. The distance is 350 ms, which is above the threshold. Hence, $P_{3}$ is considered as a systolic peak. In the next round, we choose $P_{2}$ and follow the same procedure. As the distance between $P_{2}$ and $P_{1}$ is less than the threshold, $P_{2}$ is not a systolic peak and is removed from the false-peak list.

## 6. Evaluation and Results

We evaluate the proposed method using the PPG data collected via the Samsung smartwatches in free-living conditions. The evaluation includes the data of 36 healthy individuals. The model generalization is an essential factor that should be taken into consideration. We validate the performance of the proposed method by implementing an inter-patient test, in which training and testing data are selected from separate individuals. In this regard, the PPG data of 26 participants (i.e., 9,600,000 15 s segments) are utilized for the training phase. We train the proposed model using (1) the noisy PPG signals constructed via the generator function and (2) their true labeling vectors. For the testing phase, the data of the remaining 10 participants (i.e., 35,800 15 s segments) are selected. We separate the users to avoid any data leakage between the model training and testing. Similar to the training phase, the generator function is utilized to create noisy PPG segments. The test PPG signals are fed to the model, and the labeling vectors are estimated. Then, the method’s performance is assessed by comparing the estimated labeling vectors with the true labeling vectors.

In our experiments, we used a Linux machine with AMD Ryzen Threadripper 2920X 12-Core processor, NVIDIA TITAN RTX GPU (24 GB memory), and 126 GB RAM. We used Tensorflow (v2) deep learning framework with high-level Keras API to construct our model. A batch size of 800 and 200 epochs, where the number of steps per epoch was 60, was selected for model training. In the training data, the range of SNR is from −2.5 to 47.5 dB (complete noisy to noise-free signal). The data are clustered into 10 ranges with the step of 5 dB. This balancing prevents the network from being over-learned for a specific SNR value. A total number of 9,600,000 segments were used for the training phase (90% training and 10% validation). Figure 12 indicates four examples of 15 s noisy PPG signals, with different noise levels, used in the training phase. The method was implemented using Tensorflow [57], Keras [58], and SciPy [59] in Python.

In addition to the proposed method, we implemented five exiting methods for PPG peak detection. First, Elgendi et al.’s method [60] was performed, using a dynamic threshold and two event-related moving average methods. Second, we utilized Van Gent et al.’s method [36] as an adaptive threshold method. Van Gent et al. [36] used an adaptive threshold along with a moving average on both sides of each sample. Third, Kuntamalla et al.’s [8] method was implemented to estimate PPG peaks using an adaptive threshold, which is empirically set to 0.35. Fourth, Chakraborty et al. [40], as a transform-based method, was used to estimate the peaks’ locations using a Hilbert transform. Finally, a 1D-CNN method was implemented. The model was fully convolutional, consisting of six 1D stacked convolutional layers. The input data were the same as the inputs of the proposed method. It should be noted that for the Elgendi and Van Gent methods, we used the versions implemented in Neurokit [61] and Heartpy [62] Python packages.

### 6.1. Evaluation Measures

A beat-to-beat comparison was made between the detection results and the reference test set label to evaluate the algorithm in terms of accuracy. In the comparison, true positive (TP) is when the PPG peaks are detected correctly, false negative (FN) is when the method fails to detect a peak, and false positive (FP) is when the algorithm detects, e.g., noise as a peak. Then, the performance of the proposed method was assessed by calculating precision, recall, and F1-score, as follows [39]:(7)$$\mathrm{precision}=\frac{\mathrm{TP}}{\mathrm{TP}+\mathrm{FP}}$$
(8)$$\mathrm{recall}=\frac{\mathrm{TP}}{\mathrm{TP}+\mathrm{FN}}$$
(9)$$\mathrm{F1-score}=2\times \frac{\mathrm{precision}\times \mathrm{recall}}{\mathrm{precision}+\mathrm{recall}}=2\times \frac{\mathrm{TP}}{\mathrm{TP}+\mathrm{FP}+\mathrm{FN}}$$

### 6.2. Test Set Results

Our proposed method was evaluated using the test dataset created by the generator function. The function generated 100 Hz noisy PPG signals along with the SNR values and the corresponding labeling vectors. A total of 35,800 15 s noisy PPG signals with a balanced range of SNR were used for the testing. The SNR values were between −2.5 and 47.5 dB in our evaluation. The signals were divided into 5 dB SNR groups. Then, the performance of the method was investigated for each group.

Figure 13 shows a PPG segment with different peak detection results. The SNR is 8.82 dB. The vertical dash lines show the true peaks, and the markers indicate the estimated peaks. Our method was successful at locating systolic peaks in 15 s segment. However, the other methods missed one or several peaks and detected false peaks as systolic peaks. The Kuntamalla method had the worst performance in this example.

The performance of the models for different SNR groups are shown in Figure 14. A quantitative comparison is also presented in Table 2. Figure 14a illustrates the methods’ precision. All the methods except the Kuntamalla method obtain equal precision value (i.e., 98%) when the SNR is above 42.5 dB. However, the precision values drop when the SNR values decrease. For example, in SNR 45 to 25 dB, the precision for the proposed method, Elgendi, Van Gent, Chakraborty, Kuntamalla, and 1D-CNN decrease by 15%, 18%, 24%, 19%, 20%, and 16%, respectively. As indicated, the proposed deep-learning-based method outperforms the existing methods. The results show that the false positive in the proposed method is lower compared to the other methods. Therefore, our method detects fewer false peaks as systolic peaks in the noisy PPG signals.

Figure 14b indicates the methods’ recall values. The figure shows that all the methods perform well in noise-free conditions, i.e., almost 96% recall. Our method showed better recall values in all SNR ranges except in 20 dB SNR, in which our method achieved 79% while the 1D-CNN method achieved 80%. As indicated, there are decreasing trends in the recall values when the SNR decreases. The falling trends are more intense at lower SNRs. With the lowest SNR, the difference between our method and the other methods reaches the highest value. As presented in Table 2, at SNR 0 dB, the differences between our method and the other methods are 7% (Elgendi), 8% (Van Gent), 12% ( Kuntamalla), 18% (Chakraborty), and 1% (1D-CNN). The recall values show that our method obtains lower false negatives compared to the other methods. Therefore, our method is relatively more successful in detecting the true peaks.

Finally, the methods’ F1-score values are illustrated in Figure 14c. When the SNR values drop, the F1-score of the proposed method decreases with a smaller slope compared to the existing methods. The difference is bigger with lower SNR values. For example, as shown in Table 2, the F1-score values at 0 dB SNR are 0.52, 0.46, 0.43, 0.40, 0.38, and 0.48 for our method, Elgendi, Van Gent, Chakraborty, Kuntamalla, and 1D-CNN, respectively. In summary, our method was compared with five existing PPG peak detection methods using different noise levels (i.e., from noise-free PPG signals to distorted PPG signals due to high level of noise). Our findings showed that the performance of all the methods was similar in noise-free conditions, i.e., when the SNRs were higher than 40 dB. The noise-free signals included no or a few false peaks, and the methods could detect the systolic peaks successfully. However, in real-life situations, the PPG signals, collected by wearable devices, might be distorted due to motion artifact or environmental noise. With the medium-noise-level signals (i.e., SNRs from 40 to 15 dB), our method outperformed the state-of-the-art methods. With the high-noise-level signals (i.e., SNRs lower than 15 dB), the performances of all the methods (including the proposed method) were diminished. However, the results showed that our method was more successful with low-quality signals as well. Consequently, the proposed method has the best performance in all the SNR groups, particularly when the SNR values are not large. The method is more robust against noise and could better discriminate between the systolic and noise peaks.

**Computation time:** In addition to the accuracy assessment, we evaluated the computation time of the testing phase. We repeated the experiments 100 times and calculated the computation time of the methods. The average values and standard errors are indicated in Table 3. The Elgendi method (including rule-based steps) had the lowest execution time, i.e., 0.75 ms. The execution time of the proposed deep learning method was 1.081 ms, on average, which is lower than the processing time of the Van Gent, Chakraborty, Kuntamalla, and 1D-CNN methods (i.e., 8.55 ms, 2.48 ms, 2.55 ms, and 1.22, respectively).

### 6.3. Limitations and Future Work

The dataset used in this paper was limited to healthy participants. However, other studies [41] indicated that arrhythmias—such as premature atrial contraction, premature ventricle contraction, and atrial fibrillation—might affect the accuracy of peak detection methods. The method’s performance should be investigated with the data of non-healthy individuals to address the lack of generalizability of the results.

Moreover, our evaluation was restricted to one dataset collected during free-living conditions using Samsung Gear Sport smartwatches. The method performance should be evaluated with different physical activities. In our future work, we intend to validate our method with other databases, such as [63,64], in which the users are engaged more in intense physical activities such as cycling and running.

## 7. Conclusions

In this paper, we presented a robust CNN-based peak detection for PPG signals with different noise levels. The proposed method included three phases. A generator function was introduced in the first phase, combining the PPG records with different noise levels. In the second phase, a dilated CNN was proposed. The use of dilated convolutions provided a large receptive field, which enhanced the efficiency of time series processing with CNNs. In the third phase, a wrapper function was implemented to detect the location of the PPG signals. After predicting the peaks, a filtering function was used to remove the false peaks. We evaluated the proposed method using the PPG data collected via wearable devices under free-living conditions. Our method was compared with five existing PPG peak detection methods. The performances of the methods were similar with noise-free PPG. However, our method exhibited higher accuracy when the noise level increased. We showed that the average F1-score of the proposed method was 81%, while Elgendi, Van Gent, Chakraborty, Kuntamalla, and 1D-CNN methods obtained 77%, 74%, 77%, 69%, and 79%, respectively. Our results indicated that the proposed PPG peak detection method was more successful in terms of recall and precision in a noisy environment.

## Figures and Tables

**Figure 1 sensors-22-06054-f001:**
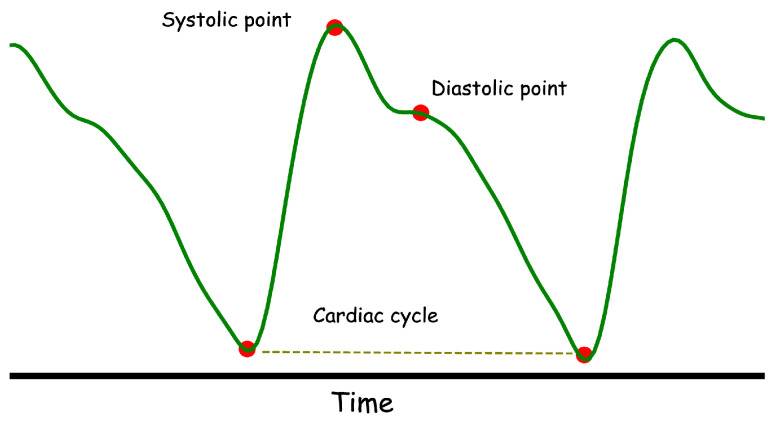
An example of a filtered PPG signal.

**Figure 2 sensors-22-06054-f002:**
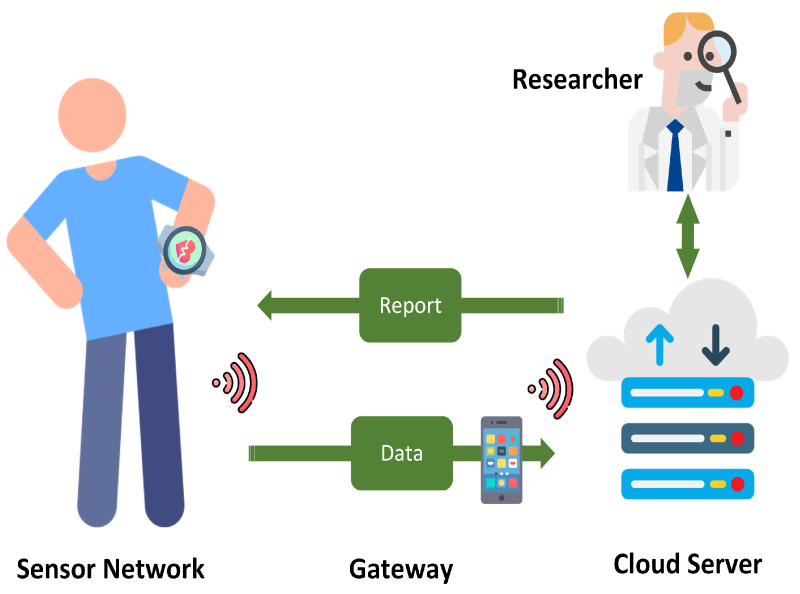
The monitoring system used for PPG collection.

**Figure 3 sensors-22-06054-f003:**
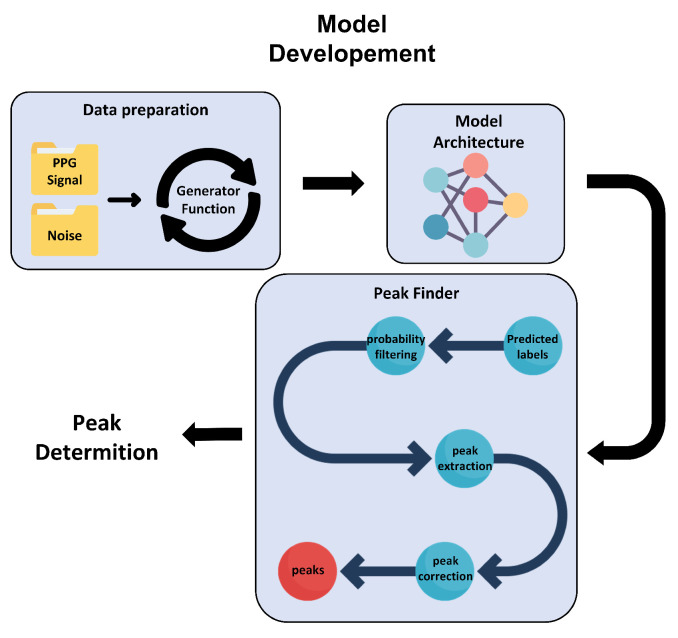
The proposed PPG peak detection method including data preparation, model architecture, and peak finder.

**Figure 4 sensors-22-06054-f004:**
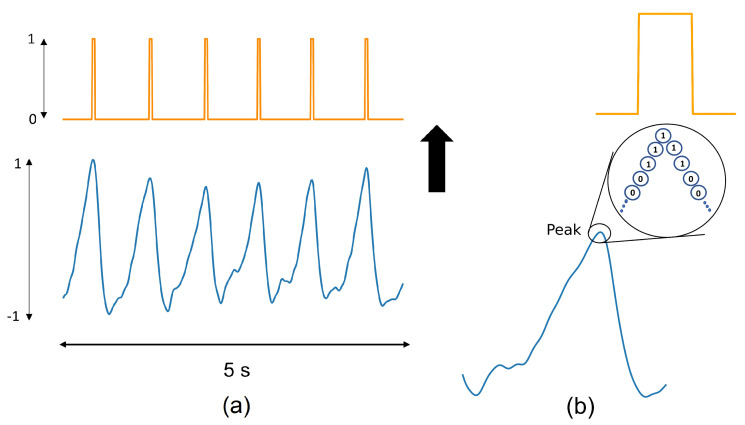
Schematic of labeling for PPG signal. (**a**) A 5 s PPG signal and its labeling vector; “1” indicates the peak’s location. (**b**) Five labels are set to “1” for each peak.

**Figure 5 sensors-22-06054-f005:**
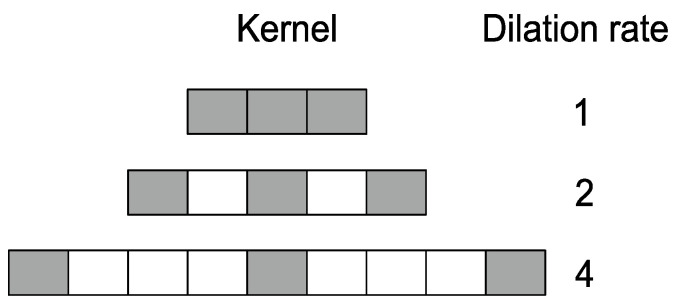
Receptive field of the kernel increases when dilation rate is increased. Grey denotes kernel weights while white indicates skipped inputs.

**Figure 6 sensors-22-06054-f006:**
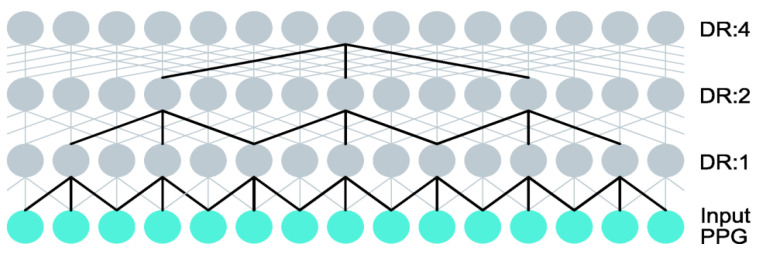
The receptive field of a neuron in a three-layer dilated convolution network is illustrated with bold lines. Note how the dilation rate (DR) is doubled at every layer. Our model is deeper than this illustration as it contains four additional layers.

**Figure 7 sensors-22-06054-f007:**
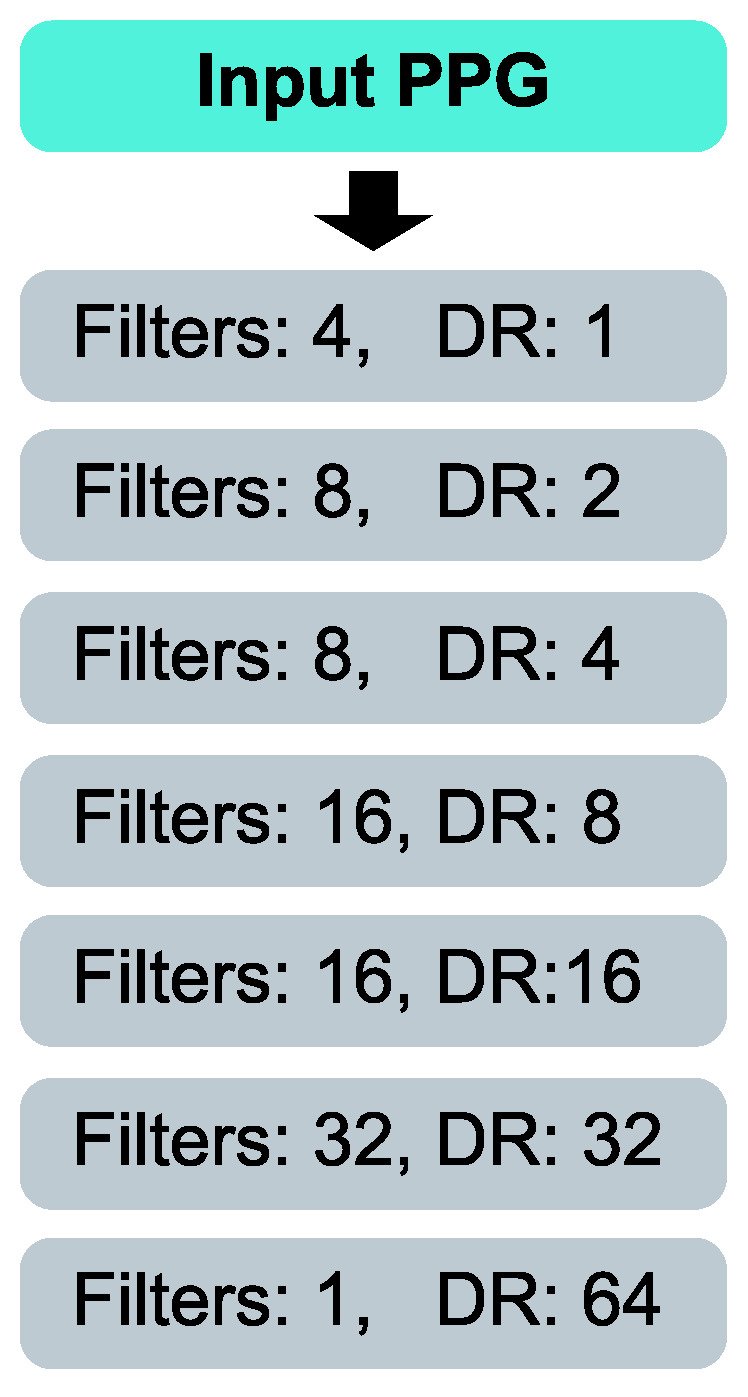
Our model is a fully convolutional neural network with seven layers. Dilatation rate (DR) is doubled at every layer, while the number of filters is slowly increased with depth.

**Figure 8 sensors-22-06054-f008:**
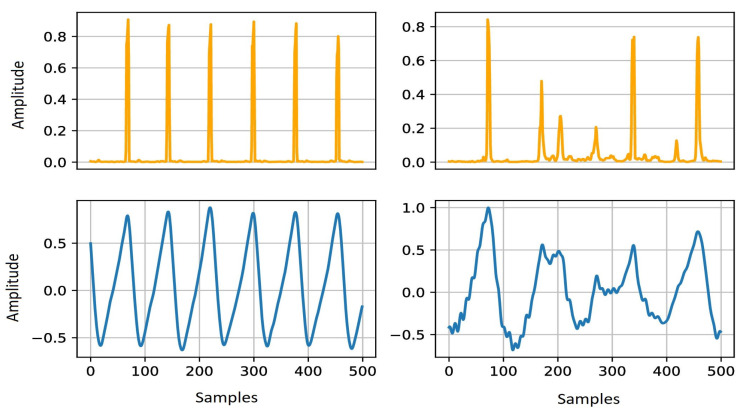
Two examples of inputs (i.e., PPG with different noise levels) and the model predictions. The lower row shows the two inputs, and the upper row includes the two labeling vectors predicted by the model.

**Figure 9 sensors-22-06054-f009:**
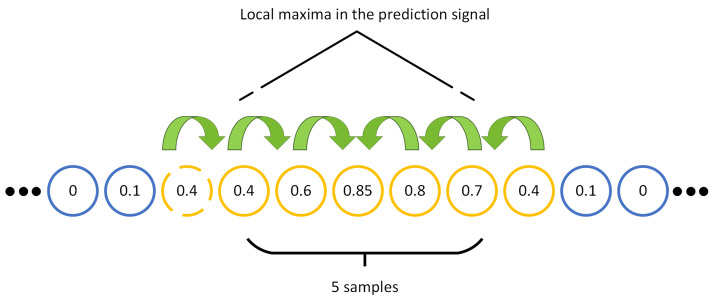
The procedure of selecting predictions that are above the given threshold.

**Figure 10 sensors-22-06054-f010:**
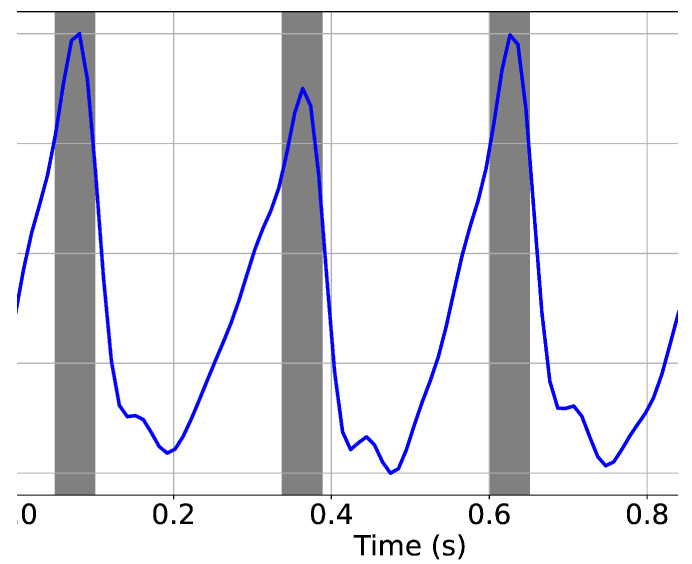
Tolerance distance (50 ms) is shaded in the figure. If the peak is detected within this range it is considered as true peak.

**Figure 11 sensors-22-06054-f011:**
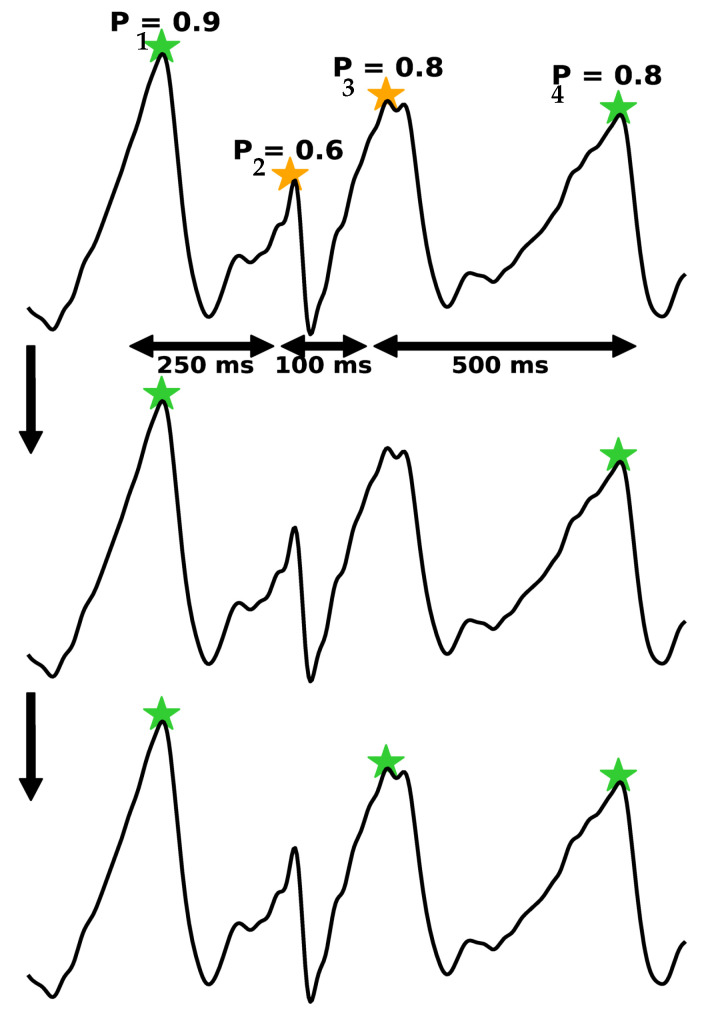
Filtering the unnaturally close peaks in the peak correction step. The upper figure shows all of the detected peaks. The middle plot illustrates the removal of the peaks that are within the threshold distance. The lowermost figure shows that the peak with the highest probability is added back into the final set of accepted peaks.

**Figure 12 sensors-22-06054-f012:**
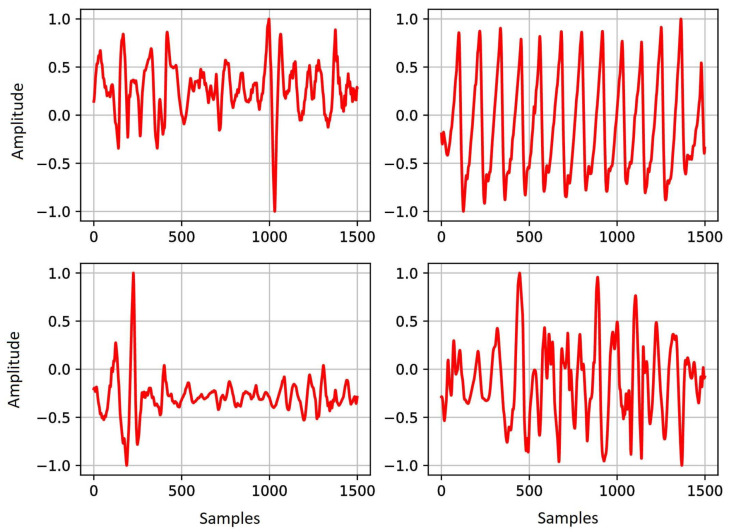
Examples of PPG data with different noise levels used in the training phase. The upper right figure illustrates a low-noise PPG signal, while the other examples contain different types of noises, such as motion artifact and baseline wander. The *x*-axis is the samples and the *y*-axis is the amplitude.

**Figure 13 sensors-22-06054-f013:**
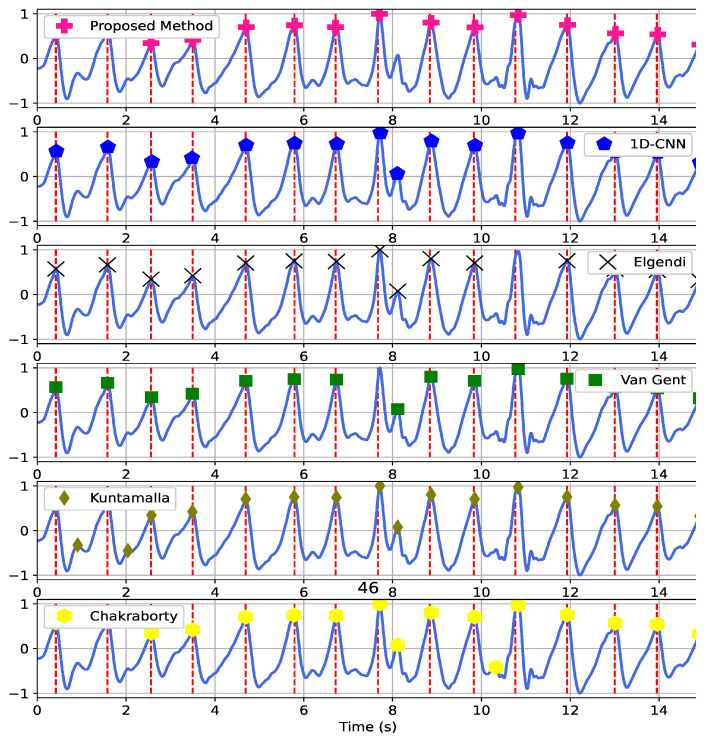
A PPG signal segment with 8.82 dB SNR and the peak detection results obtained by the methods. The vertical dash lines are true peak. The markers show the positions of the peaks detected by the methods.

**Figure 14 sensors-22-06054-f014:**
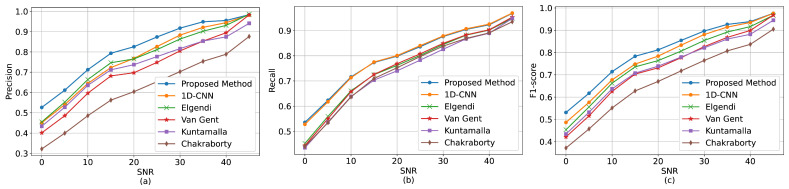
Performance comparison between the methods. (**a**) Precision, (**b**) recall, and (**c**) F1-score at different noise levels.

**Table 1 sensors-22-06054-t001:** Background information of the participants.

Characteristic	Type	Values
**Participants no.**	Men	17
	Women	19
**Age (years), mean (SD)**	Men (17)	33.5 (6.5)
	Women (19)	31 (6.8)
**BMI, mean (SD)**	Men	25.58 (2.94)
	Women	24.32 (6.17)
**Exercise**	Almost daily	9
	A few times a week	19
	Once a week or fewer	7
**Education**	Primary school	1
	High school	7
	College	8
	University	20
**Employment status**	Working	27
	Unemployed	1
	Student	6
	Other	1

**Table 2 sensors-22-06054-t002:** Performance comparison between the proposed method and the other existing methods.

Mean	Proposed Method			Elgendi			Van Gent			Chakraborty			Kuntamalla			1D-CNN		
SNR (dB)	prec.	recall	F-1	prec.	recall	F-1	prec.	recall	F-1	prec.	recall	F-1	prec.	recall	F-1	prec.	recall	F-1
45	**0.98**	**0.97**	**0.97**	**0.98**	0.94	0.96	**0.98**	0.95	0.96	**0.98**	0.94	0.96	0.87	0.92	0.90	**0.98**	0.96	0.97
40	**0.95**	**0.92**	**0.94**	0.93	0.90	0.92	0.91	0.91	0.91	0.92	0.88	0.90	0.82	0.88	0.85	0.94	**0.92**	0.93
35	**0.95**	**0.91**	**0.92**	0.90	0.88	0.89	0.86	0.88	0.87	0.88	0.83	0.85	0.78	0.85	0.81	0.92	0.90	0.91
30	**0.91**	**0.88**	**0.89**	0.87	0.85	0.86	0.81	0.85	0.83	0.85	0.79	0.82	0.74	0.82	0.78	0.88	0.87	0.88
25	**0.87**	**0.84**	**0.85**	0.80	0.79	0.80	0.74	0.80	0.77	0.79	0.73	0.76	0.69	0.77	0.73	0.82	0.83	0.83
20	**0.82**	0.79	**0.81**	0.76	0.75	0.76	0.69	0.76	0.72	0.75	0.68	0.71	0.63	0.73	0.67	0.76	**0.80**	0.78
15	**0.79**	**0.77**	**0.78**	0.75	0.73	0.74	0.68	0.73	0.71	0.75	0.66	0.70	0.60	0.69	0.64	0.72	0.77	0.74
10	**0.71**	**0.72**	**0.71**	0.66	0.66	0.66	0.60	0.66	0.63	0.67	0.58	0.62	0.52	0.61	0.56	0.64	0.71	0.68
5	**0.60**	**0.62**	**0.61**	0.55	0.56	0.56	0.49	0.55	0.52	0.55	0.46	0.50	0.42	0.50	0.46	0.53	0.61	0.57
0	**0.52**	**0.53**	**0.53**	0.46	0.46	0.46	0.42	0.45	0.43	0.46	0.35	0.40	0.35	0.41	0.38	0.45	0.52	0.48
Overall	**0.82**	**0.80**	**0.81**	0.78	0.76	0.77	0.72	0.77	0.74	0.78	0.76	0.77	0.65	0.73	0.69	0.78	0.80	0.79

(1) The corresponding PPG records with the highest precision, recall, or F1-scores (in each row) are presented in bold type. (2) The number of signals analyzed for each SNR range are 3580.

**Table 3 sensors-22-06054-t003:** Average processing time comparison of the proposed method and other existing methods.

Method	Proposed Method	Elgendi	Van Gent	Chakraborty	Kuntamalla	1D-CNN
Time (ms)	1.081 ± 0.31	0.84 ± 0.31	8.55 ± 0.73	2.48 ± 0.50	2.55 ± 0.53	1.22 ± 0.23

## Data Availability

Data sharing not applicable.

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
