# Peer review of "Robust PPG Peak Detection Using Dilated Convolutional Neural Networks"

_sensors, 2022, doi:10.3390/s22166054_

Round 1

Reviewer 1 Report

This study developed a new peak detection algorithm for PPG signals using dilated convolutions CNN and post-processing discarding false peaks. With some minor modifications, the paper would bring academic interest to readers.

1. In Table 1: Please provide the number of men and women included as participants.

2. In line 274: Which method is used for upsampling?

3. typo

In line 441: P2

In line 515: 0.79%, 0.80%

Reviewer 2 Report

The article "Robust PPG peak detection using dilated convolutional neural networks" by Kianoosh Kazemi describes neural network-based method to detect peaks on noizy PPG readings from the sensors. The overall article descign looks scientifically sound and the article can be accepted after the minor revision. The main problems that should be addressed include:

* what he problem with the modern state of the art solutions. The method demonstrate the same performance metrics as the best modern solution - why and where it can outperform them significantly to be important for the users of PPG technology.

* The influence of noize is investigated, but more information should be provided concerning what is the typical noize levels and in which cases (what % of overall cases) the proposed method outperform the best solution and provide more reliable result

* The review part looks very short for such common task and probably should be extended.
